# Biological Role of Tumor/Stromal CXCR4-CXCL12-CXCR7 in MITO16A/MaNGO-OV2 Advanced Ovarian Cancer Patients

**DOI:** 10.3390/cancers14071849

**Published:** 2022-04-06

**Authors:** Crescenzo D’Alterio, Anna Spina, Laura Arenare, Paolo Chiodini, Maria Napolitano, Francesca Galdiero, Luigi Portella, Vittorio Simeon, Simona Signoriello, Francesco Raspagliesi, Domenica Lorusso, Carmela Pisano, Nicoletta Colombo, Gian Franco Zannoni, Nunzia Simona Losito, Rossella De Cecio, Giosuè Scognamiglio, Daniela Califano, Daniela Russo, Valentina Tuninetti, Maria Carmela Piccirillo, Piera Gargiulo, Francesco Perrone, Sandro Pignata, Stefania Scala

**Affiliations:** 1Microenvironment Molecular Targets Unit, Istituto Nazionale Tumori IRCCS—Fondazione G. Pascale, 80131 Napoli, Italy; c.dalterio@istitutotumori.na.it (C.D.); a.spina@istitutotumori.na.it (A.S.); m.napolitano@istitutotumori.na.it (M.N.); f.galdiero@istitutotumori.na.it (F.G.); l.portella@istitutotumori.na.it (L.P.); d.califano@istitutotumori.na.it (D.C.); d.russo@istitutotumori.na.it (D.R.); 2Clinical Trials Unit, Istituto Nazionale Tumori IRCCS—Fondazione G. Pascale, 80131 Napoli, Italy; l.arenare@istitutotumori.na.it (L.A.); m.piccirillo@istitutotumori.na.it (M.C.P.); piera.gargiulo@istitutotumori.na.it (P.G.); f.perrone@istitutotumori.na.it (F.P.); 3Section of Statistics, Department of Mental Health and Public Medicine, Università degli Studi della Campania Luigi Vanvitelli, 80138 Napoli, Italy; paolo.chiodini@unicampania.it (P.C.); vittorio.simeon@unicampania.it (V.S.); simona.signoriello@unicampania.it (S.S.); 4Gynecological Oncology Unit, Fondazione IRCCS Istituto Nazionale dei Tumori, 20133 Milan, Italy; francesco.raspagliesi@istitutotumori.mi.it; 5Division of Gynecologic Oncology, Department of Women and Child Health, Fondazione Policlinico Universitario A. Gemelli IRCCS, 00168 Rome, Italy; domenica.lorusso@policlinicogemelli.it; 6Department of Life Science and Public Health, Catholic University of Sacred Heart, Largo Agostino Gemelli, 00168 Rome, Italy; 7Urogynaecological Medical Oncology, Istituto Nazionale Tumori IRCCS—Fondazione G. Pascale, 80131 Napoli, Italy; c.pisano@istitutotumori.na.it (C.P.); s.pignata@istitutotumori.na.it (S.P.); 8Gynecologic Cancer Program, Università degli Studi di Milano-Bicocca, 20126 Milan, Italy; nicoletta.colombo@unimib.it; 9Gynecopathology and Breast Pathology Unit, Department of Woman, Child and Public Health Sciences, Fondazione Policlinico Universitario A. Gemelli IRCCS, 00168 Rome, Italy; gianfranco.zannoni@unicatt.it; 10Pathological Anatomy Institute, Catholic University of Sacred Hearth, 00168 Rome, Italy; 11Pathology Unit, Istituto Nazionale Tumori IRCCS—Fondazione G. Pascale, 80131 Napoli, Italy; n.losito@istitutotumori.na.it (N.S.L.); r.dececio@istitutotumori.na.it (R.D.C.); giosue.scognamiglio@istitutotumori.na.it (G.S.); 12FPO IRCCS Istituto di Candiolo, 10060 Turin, Italy; valentina.tuninetti@ircc.it

**Keywords:** chemokine, CXCR4-CXCL12-CXCR7 axis, ovarian cancer, prognosis, tumor microenvironment

## Abstract

**Simple Summary:**

Despite rapid progress in the research on epithelial ovarian cancer (EOC), it is usually diagnosed during the advanced stage with only 30% of patients surviving longer than 5 years. This is the first study in which the whole CXCR4-CXCL12-CXCR7 axis was systematically evaluated in tumor and stromal cells, through rigorous statistical methods in a prospective clinical trial. CXCL12 expression in cancer cells is associated with worse progression-free survival in stage III EOC patients, and deserves further attention as a potential prognostic and therapeutic target.

**Abstract:**

This study investigated the prognostic role of the CXCR4-CXCL12-CXCR7 axis in advanced epithelial ovarian cancer (EOC) patients receiving first-line treatment within the MITO16A/MaNGO-OV2 phase-IV trial. CXCR4-CXCL12-CXCR7 expression was evaluated in the epithelial and stromal component of 308 EOC IHC-stained tumor samples. The statistical analysis focused on biomarkers’ expression, their association with other variables and prognostic value. Zero-inflated tests, shrinkage, bootstrap procedures, and multivariable models were applied. The majority of EOC (75.0%) expressed CXCR4 and CXCR7, 56.5% expressed the entire CXCR4-CXCL12-CXCR7 axis, while only 4.6% were negative for CXCL12 and its cognate receptors, in regard to the epithelial component. Stromal CXCL12 and CXCR7, expressed in 11.2% and 65.5%, respectively, were associated with the FIGO stage. High CXCL12 in epithelial cancer cells was associated with shorter progression-free and overall survival. However, after adjusting for overfitting due to best cut-off multiplicity testing, the significance was lost. This is a wide-ranging, prospective study in which CXCR4-CXCL12-CXCR7 were systematically evaluated in epithelial and stromal components, in selected stage III-IV EOC. Although CXCL12 was not prognostic, epithelial expression identified high-risk FIGO stage III patients for PFS. These data suggest that it might be worth studying the CXCL12 axis as a therapeutic target to improve treatment efficacy in EOC patients.

## 1. Introduction

Epithelial ovarian cancer (EOC) accounts for approximately 90% of ovarian cancers [1]. The majority (60%) are high-grade serous carcinomas (HGSCs), followed by endometrial, clear cell, mucinous, and low-grade serous carcinomas [2]. Cytoreductive (debulking) surgery and platinum/paclitaxel chemotherapy are the standard therapeutic approaches [1]. While stage I and II EOC patients have a 76–91% 5-year survival rate [3], only 30% of patients with advanced-stage disease survive more than 5 years [1]. Poor prognosis depends on late diagnosis, acquired resistance to platinum-based regimen [4] and recurrence [5]. The anti-VEGF antibody, bevacizumab, prolongs progression-free survival (PFS) in newly diagnosed and relapsed EOCs, and in combination with carboplatin plus paclitaxel is used for maintenance in first-line therapy [6]. Recently, poly (adenosine diphosphate-ribose) polymerase (PARP) inhibitors were introduced into the treatment of first-line and relapsed ovarian cancer [7,8]. Thus, it is crucial to identify the biological features to allow for patient characterization and the optimal therapeutic approach [9].

CXCR4 is a chemokine receptor that induces cell chemotaxis following a CXCL12 gradient [10]. CXCR4 participates in the EOC development [11], is expressed by ovarian cancer, tumor and stromal cells (myeloid or T cells), and cooperates with angiogenic factors to generate new vessels [12]. Silencing CXCR4 in human ovarian cancer cells reduces cell proliferation, migration and invasion and significantly decreases in vivo tumor development [11,13]. The only FDA-approved CXCR4 antagonist, AMD3100, sensitizes ovarian cancer to chemotherapy [14] and impairs the epithelial–mesenchymal transition (EMT) when coupled with paclitaxel (PTX)-loaded bovine serum albumin (BSA) nanoparticles [15,16]. CXCR4 further plays a role in cisplatin-resistance inducing let-7a, which suppresses the pro-apoptotic BCL-XL/S [17]. CXCL12 is present in 95% of ovarian cancer ascites, while CXCR4 is the only chemokine receptor expressed in ovarian cancer cells [18]. VEGF and CXCL12 cooperate to induce potent neovascularization, as VEGF up-regulates CXCR4 on vascular endothelial cells and synergize CXCL12-mediated vascular endothelial cells migration [19] through matrix metalloproteinases (MMPs) [20], while CXCR4 is overexpressed in high-grade serous carcinomas [12] and correlates with stage and metastasis development [10,21]. CXCL12 is an independent predictor of poor survival in ovarian cancer [22]. CXCR7/ACKR3 is the second CXCL12 receptor, and is also activated by interferon-inducible T-cell alpha chemoattractant I-TAC (CXCL11) [23]. The role of CXCR7 in EOC is poorly characterized, although the ligands, CXCL11 and CXCL12, are overexpressed in ovarian carcinomas [16,24]. In this study, we evaluated the epithelial and stromal cellular expression of CXCR4-CXCL12-CXCR7 in tumors from advanced ovarian cancer patients enrolled in the MITO16A/MaNGO-OV2 trial [25]. The MITO16A trial is a multicenter, phase IV, single arm trial. The aim of the trial was to explore the clinical and biological prognostic factors for advanced ovarian cancer patients receiving first-line treatment with carboplatin, paclitaxel, and bevacizumab, Patients with advanced (stage IIIB-IV) or recurrent, previously untreated, ovarian cancer received carboplatin, paclitaxel plus bevacizumab for six 3-weekly cycles followed by bevacizumab single agent until progression or unacceptable toxicity up to a maximum of 22 total cycles. Both progression-free and overall survival were used as endpoints. After a median follow-up of 32.3 months (IQR 24.1–40.4), the median progression-free survival was 20.8 months (95% CI 19.1 to 22.0) and median overall survival was 41.1 months (95% CI 39.1 to 43.5). Performance status, stage, and residual disease after primary surgery were the most important clinical prognostic factors [25]; in this context, the prognostic role of epithelial and stromal expression of CXCR4-CXCL12-CXCR7 was evaluated.

## 2. Materials and Methods

### 2.1. Description of the Study

The MITO16A/MaNGO-OV2 (www.clinicaltrials.gov (accessed on 31 March 2022) number: NCT01706120 or EudraCT number: 2012-003043-29, hereafter indicated as MITO16A) is a single arm, multicenter, open-label, non-comparative, phase IV trial of first-line chemotherapy (carboplatin plus paclitaxel) plus bevacizumab, followed by bevacizumab as single agent until progression or up to 22 total cycles. Overall, 398 patients were prospectively enrolled from 47 participant centers. Twelve research groups designed the trial as an exploratory study and no a priori hypothesis was defined to calculate the sample size of the trial. The primary aim of the MITO16A was to study clinical and biological factors for their value as prognostic factors for progression-free or overall survival among patients receiving chemotherapy and bevacizumab as a first-line treatment.

Data were collected through https://usc-intnapoli.net (accessed on 31 March 2022) website. All participants signed an informed consent for both the clinical and translational part before entering the study.

### 2.2. Sample Collection and Review

Formalin-fixed, paraffin-embedded (FFPE) tumor blocks were collected and stored according to Italian guidelines [26] and pathological revision was performed at the Instituto Nazionale Tumori of Naples IRCCS “G. Pascale” as previously reported [27].

### 2.3. Preparation of Tissue Microarray

Using the most representative areas (at least 50% of cancer cells without necrosis) from each FFPE sample, a tissue microarray (TMA) was built. Three cores (1 mm) were collected from each of the eligible tumor blocks of 311 patients and arrayed into a recipient paraffin block (35 mm × 20 mm) using a semiautomatic tissue array instrument (Galileo CK3500 TMA, ISENET, Milan, Italy). Eventually, 7 TMA blocks were built and cut into 4 μm thick sections. The sample adequacy was evaluated.

### 2.4. Immunohistochemistry (IHC)

TMA sections were deparaffinized and rehydrated. Antigen retrieval was achieved in 10 mM sodium citrate buffer at pH 6.0, using microwave for 3 cycles of 10 min at 1-fold 2 min at 600 W and 2-fold 4 min at 360 W each. Slides were treated with 3% hydrogen peroxide for 10 min at 24 °C to stop endogenous peroxidase, and then blocked in biotin/avidin solution for 10 min. Immunostaining was performed by incubating with anti-CXCR4 (1:100, MAB172, R&D systems) or anti-CXCR7 (1:70, MAB42273, R&D systems) or anti CXCL12 (1:50, MAB350, R&D systems) at 4 °C overnight. Anti-mouse secondary antibodies and peroxidase-labeled streptavidin were used (Dako, Carpinteria, California, CA, USA); diaminobenzidine reagent was employed to detect signals. The staining intensity was scored as 0 (negative), 1 (weak), 2 (moderate), or 3 (strong). The percentage of cells stained was counted. The H-Score was calculated [22]. The markers’ expression was separately evaluated on epithelial ovarian cancer cells and stromal cells within each TMA core.

### 2.5. Statistical Analyses

Data were described using the mean and standard deviation (SD) or median and interquartile range (IQR) for quantitative variables, and absolute and relative frequency for categorical variables. Each single biomarker was graphically described by histogram, evaluating its distribution and highlighting the presence of high-frequencies of 0 values. The correlation between biomarkers was investigated graphically by scatterplot and by a modified version of the Kendall test for zero-inflated values [28]. The association of each biomarker with the clinical prognostic factors was assessed using the Wilcoxon rank test for zero-inflated data (ZIW) for dichotomous variables and the Kruskal–Wallis test for zero-inflated data (ZIKW) for categorical variables, both with the use of a permutation test. Within each box plot, a circle was added at the bottom and its size was proportional to the percentage of 0 values. The prognostic effect of the clinical characteristics and biomarkers was tested considering progression-free survival (PFS) and overall survival (OS) as endpoints. PFS was defined as the time elapsing from inclusion in the study to the first occurrence of disease progression or death from any cause. OS was defined as the time elapsing from inclusion in the study and death from any cause.

To adjust the biomarker analyses, a prognostic model that included only clinical covariates was developed that was consistent with previously reported analyses (24). The model included: age (as category <65 vs. ≥65), ECOG performance status (PS 0 vs. PS 1–2), residual disease (none; ≤1 cm; >1 cm; not operated), FIGO stage (III vs. IV) and tumor histology (high-grade serous vs. other).

For each biomarker, tumor and stromal expression were paired and studied using univariate and multivariable analyses. First, for each biomarker, a model was estimated with the H-index of epithelial and stromal cells as a continuous variable after testing the linearity assumption using fractional polynomial and a dummy variable to account for 0 value. The interaction between tumor and stromal biomarker expression was also tested within this model. Second, the best cut-off value for each biomarker component (tumor or stroma) was defined as the one that minimizes the *p*-value of HR in a model adjusted for the other component (tumor or stroma); the best cut-off search was calculated on PFS (considered as the most sensitive end-point due to a higher number of events) and then applied to the OS. The OS and PFS curves for each biomarker category according to the identified cut-offs were estimated with the Kaplan–Meier method, and compared with a two-sided log-rank test.

Third, in the multivariable analyses with the Cox proportional hazard model, the prognostic value of each biomarker pair (both as continuous and as categorical according to the previously calculated cut-off) was adjusted for the clinical characteristics (see above) and reported as hazard ratios (HRs) and 95% confidence intervals (CIs).

To adjust for overfitting, HR estimates of best cut-off categories were calculated using a shrinkage procedure with the 95% CI calculated with a bootstrap percentile method [29].

Finally, in order to generate new biological–clinical hypotheses, exploratory analyses were performed to estimate the effect of biomarker best cut-off categories for predefined subgroups of patients, defined by categories of clinical variables (tumor histology, FIGO stage and residual disease), by means of the Cox regression model and the heterogeneity of the effect was measured by the interaction test. Such analyses were only conducted with PFS as the clinical outcome because of the higher number of events.

Data were analyzed using R software version 3.6.0 (R Foundation for Statistical Computing, Vienna, Austria).

## 3. Results

### 3.1. Patients Clinical and Pathological Data

Out of 398 patients enrolled in the MITO16A study, samples were available for 308 (77%) patients. Tumor blocks from thirteen patients were not available, sixty-two samples had insufficient amounts of tumor tissue and were considered inadequate for the tissue microarray (TMA). A further 12 samples were excluded because they were collected after neoadjuvant chemotherapy. Out of 311 patients used for the TMA, 3 patients were excluded due to technical issues. Finally, 308 patients were fully characterized and the data were available for statistical analysis (Appendix A, Appendix A). Among the 308 patients, the median age was 58.8 years (range 49.8–65.9), with *n* = 88 patients (28.6%) ≥65 years old (the other age classes were 18–30: 1 (0.3%); 30–45: 31 (10.1%); 45–65: 188 (61.0%)). At diagnosis, the majority of patients had a good performance status score of PS ECOG 0 (80.2%) and FIGO stage IIIC (71.1%). According to the central histological revision, high-grade serous carcinoma (HGSC) was the most common histological type (85.7%). Patient characteristics are reported in Appendix A, Appendix A. The clinical prognostic factors were reported in Appendix A, Appendix A.

### 3.2. CXCR4-CXCL12-CXCR7 Axis Is Highly Expressed in Advanced EOC

CXCR4-CXCL12-CXCR7 expression was evaluated for the epithelial and stromal component in 308 EOC samples through IHC. CXCR4 staining was cytoplasmic, perinuclear and nuclear in both epithelial and stromal cells (Figure 1A,B).

The median CXCR4 H-score was 63.3 (interquartile range (IQR) 30.0–93.3) in epithelial and 20.0 (IQR 0.0–46.7) in stromal cells (Figure 2A). CXCL12 staining was observed in cell membranes and cytoplasm; in particular, CXCL12 showed predominantly diffuse strong epithelial membrane positivity (Figure 1C,D). The median H-index of CXCL12 was 20.0 (IQR 0.0–70.0) in epithelial cells and 0 in stromal cells, with few positive cases (11.5%) (Figure 2A).

The CXCR7 expression was mainly cytoplasmic and heterogeneous (Figure 1E,F). The median CXCR7 expression was 33.3 (IQR 10.0–70.0) in epithelial and 16.7 (IQR 0.0–43.3) in stromal cells (Figure 2A). In addition, 56.5% of the EOCs expressed the entire axis CXCR4-CXCL12-CXCR7, while only 4.5% were negative for both CXCL12 and its cognate receptors (Table 1). The majority of stromal infiltrating cells express CXCR4 and CXCR7 (40.6%) receptors (Table 1).

The correlations among the markers were weak, suggesting that CXCR4, CXCL12 and CXCR7 expression is independent in EOC (Appendix A, Appendix A). Stromal CXCR7 and CXCL12 were significantly associated with FIGO stage (*p* = 0.007 and 0.033, respectively). In particular, stromal CXCR7 and CXCL12 were significantly lower in FIGO stage IV (Figure 2B).

### 3.3. Prognostic Meaning for CXCR4-CXCL12-CXCR7 in Advanced EOC

Out of 308 patients analyzed, 221 (72%) progressed and 102 (33%) died. In the univariate survival analysis considering continuous biomarkers, no significant associations were detected for PFS and OS (Appendix A, Appendix A). In univariate analysis considering the biomarkers’ cut-off, high epithelial CXCL12 appeared to be negatively associated with PFS (HR 1.39, 95%CI 1.06–1.81, *p* = 0.016) and OS (HR 1.64, 95%CI 1.11–2.42, *p* = 0.014); no associations were found for CXCR7 and CXCR4 and PFS or OS (Table 2).

The Kaplan–Meier curves for CXCL12 on PFS are shown in Figure 3A. However, using the shrinkage procedure with bootstrap confidence intervals, in order to adjust for overfitting due to best cut-off multiplicity testing, no biomarker was significantly associated with either PFS or OS (Table 2). Similar results were found in multivariate analysis considering both best and continuous biomarkers (Appendix A, Appendix A). A significant interaction was found in the PFS analysis between epithelial CXCL12 and FIGO stage, the former being associated with worse PFS among stage III patients but not within the smaller stage IV subgroup, with a *p* value for interaction of 0.044 (Figure 3B). An analysis of the interaction of the CXCR7 and CXCR4 best cut-off values for PFS with the main variables is shown in the Appendix A, Appendix A.

Moreover, CXCR4, CXCL12 and CXCR7 expression was evaluated according to treatment response in 194 patients (63% of the study population) who represented the patients with measurable residual disease. Interestingly, higher epithelial CXCR4 expression was revealed in responders to treatment with carboplatin-paclitaxel and bevacizumab (Appendix A, Appendix A).

## 4. Discussion

Genomic and molecular characterization of ovarian cancer allows a correct therapeutic approach. To characterize the tumor/tumor microenvironment, CXCR4-CXCL12-CXCR7 were evaluated in the epithelial and stromal compartment of 308 advanced EOC patients enrolled in the MITO16A/MaNGO-OV2 trial [25,27]. The entire CXCR4-CXCL12-CXCR7 axis was independently overexpressed in ovarian epithelial cancer tissue as compared to stroma. Stromal CXCR7 and CXCL12, although expressed in a minimal percentage of patients, were significantly lower in stage IV as compared to stage III patients. The evidence that indicates low stromal CXCL12 may favor metastases was previously described in breast cancer [30]. In terms of prognosis, epithelial CXCL12 correlated with PFS and OS although the significance was lost after the shrunken bootstrap analysis cross-validation was applied to avoid cut-offs overfitting. To the best of our knowledge, among several studies [22,31,32] describing the crucial role of the CXCR4-CXCL12 axis in ovarian cancer, this is the first prospective study in which the entire axis CXCR4-CXCL12-CXCR7 has been systematically evaluated in epithelial and stromal components, in a representative (*n* = 308) population of accurately selected stage III-IV ovarian cancer patients with adequate statistical analysis. However, the evaluated cohort of patients, which was comprised of FIGO IIIB-C and IV stage patients, might not be ideal for identifying prognostic factors as the early stages of disease are missing. Our findings suggest that high CXCR4 at diagnosis might identify, within the category of patients with poor prognosis ovarian cancer (R > 0), patients that could benefit from bevacizumab treatment [19]. Further studies are ongoing to better categorize the angiogenetic pathway expression in the study population.

Although the intent of the MITO16A/MaNGO-OV2 trial was “to explore the clinical and biological prognostic factors for advanced ovarian cancer patients receiving first-line treatment with carboplatin, paclitaxel, and bevacizumab”, data on the feasibility of interval debulking surgery (IDS) after carboplatin–paclitaxel–bevacizumab [33] and in the absence of relevant thromboembolic events and antithrombotic prophylaxis in advanced ovarian cancer patients treated with bevacizumab [34] were also reported. Recently, the first molecular results of the biomarkers’ evaluation in MITO16A/MaNGO-OV2 patients was reported. Seven angiogenesis-related proteins and twelve microRNAs’ angiosignatures were evaluated, demonstrating that high miR-484, a VEGFB-targeting miRNA, was associated with longer progression-free and overall survival although the shrinkage procedure to adjust for over-fitting hazard ratio estimates reduced the association significance [35]. Previous retrospective studies have reported the prognostic value of CXCL12 in 289 ovarian cancer patient as evaluated through tissue microarray [22], while a small study (44 EOC with a median 37 months follow-up) showed no prognostic meaning for CXCL12 [10]. Due to the low number of tumor infiltrating lymphocytes (TIL) and low PD-L1, ovarian cancer TME is classified as “cold”, although the tumor mutational burden is relatively high [35]. Nevertheless, TME varies according to the histological subtypes as the majority of HGSOC (83%) present CD8 + TILs, while low-grade serous and endometrioid carcinoma had lower TILs and intra-epithelial CD8 + T cells associated with prolonged survival [36]. Neoadjuvant chemotherapy also affects TME through higher cytolysis activity, natural killer (NK) infiltration and the expansion of T cells [37], conversely increasing expression of the immune checkpoints PD1, PD-L1 and CTLA-4 [38]. In ovarian cancer TME, CXCL12 is secreted by several cells [39,40] and drives Tregs recruitment, inhibits leukocytes and facilitates M2 macrophage polarization, driving the TME toward an immunosuppressive status [41]. Interestingly, the most immunosuppressive CAF-S1 in EOC is rich in CXCL12 (CXCL12β) [42]. CXCL12–CAF induced EMT through the CXCR4/Wnt/β-catenin pathway, promoting cell proliferation and cisplatin resistance. Stromal CXCL12 is an independent prognostic marker of platinum sensitivity [43].

Based on the role played by CXCL12 in the EOC epithelia and stroma, targeting the CXCR4-CXCL12 axis has been coupled with immune checkpoint inhibitors antibodies, as the efficacy of single-agent immune-checkpoint blockade in EOC is limited [44].

The CXCR4 antagonist, AMD3100 and anti-PD-1 significantly enhanced antitumor effects by potentiating effector T-cell infiltration, function and increasing memory T cells in “in vivo” models. AMD3100 promoted a reduction in intra-tumoral Tregs and the conversion of Treg cells into T-helpers [41]. In a syngeneic ovarian cancer model, locoregional delivery of the oncolytic vaccinia virus (OVV)-plus CXCR4 antagonist (CXCR4-A) reduced the tumor load and promoted the immune response through CD103 + dendritic cells [45]. Thus, we can speculate that EOC patients might benefit from CXCL12 axis blockade in combination with chemotherapy, immunotherapy and molecular therapies targeting macrophages, Tregs and fibroblasts [35].

## 5. Conclusions

Our findings shed light on the key role of CXCL12, CXCR7 and CXCR4 in EOC. In univariate analysis, high epithelial CXCL12 appeared to be negatively associated with PFS and OS, although after adjusting for overfitting the significance was lost. A significant interaction was detected in the PFS analysis between epithelial CXCL12 and FIGO III stage patients. Thus, although no prognostic factors were identified through objective and rigorous statistical methods with ovarian cancer clinical data, the data support the rationale for CXCL12 targeting therapy in EOC management.

## Figures and Tables

**Figure 1 cancers-14-01849-f001:**
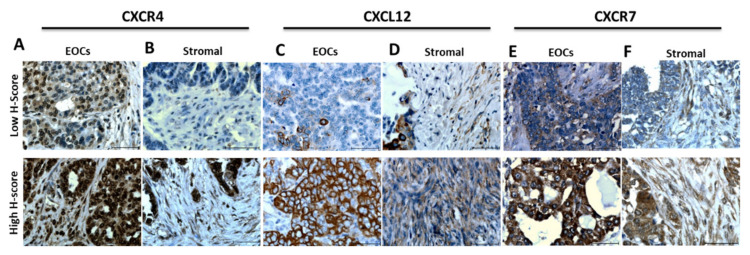
CXCR4-CXCL12-CXCR7 axis is expressed in epithelial and stroma in advanced EOC. CXCR4-CXCL12-CXCR7 expression in epithelial (**A**,**C**,**E**) and stromal cells (**B**,**D**,**F**) for low (upper panel) and high H-score (lower panel) expression based on best cut-off value for each biomarker component (tumor or stroma) (400× magnification; bars 50 µm).

**Figure 2 cancers-14-01849-f002:**
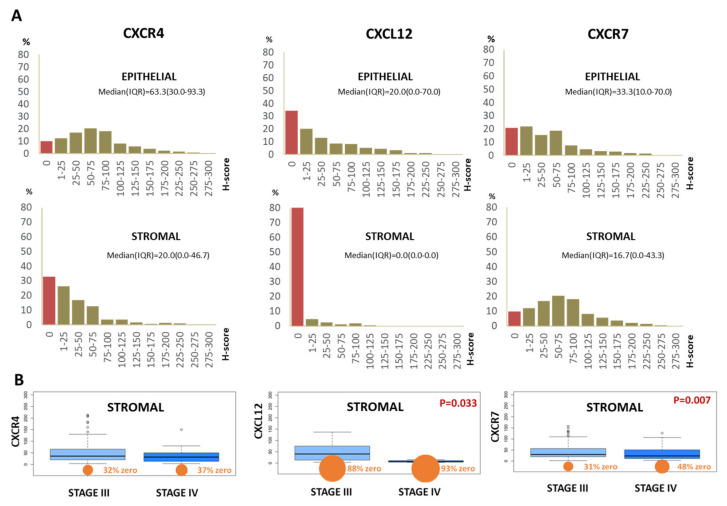
Decrease in stromal CXCR7 and CXCL12 in FIGO stage IV. Relative distribution of CXCR4-CXCL12-CXCR7 in EOC epithelial and stroma (**A**). CXCR4-CXCL12-CXCR7 percentage in epithelial and stromal cells (% of epithelial) in each H-score interval (X axes). Y axes showed the % of individuals with a given interval of CXCR4-CXCL12-CXCR7 expression. The median with interquartile range (IQR) was also reported, as a measure of central tendency for a skewed dataset. Box whisker plots showed the association between FIGO stage (stage III vs. IV) and CXCR4- CXCL12-CXCR7 expression (**B**).

**Figure 3 cancers-14-01849-f003:**
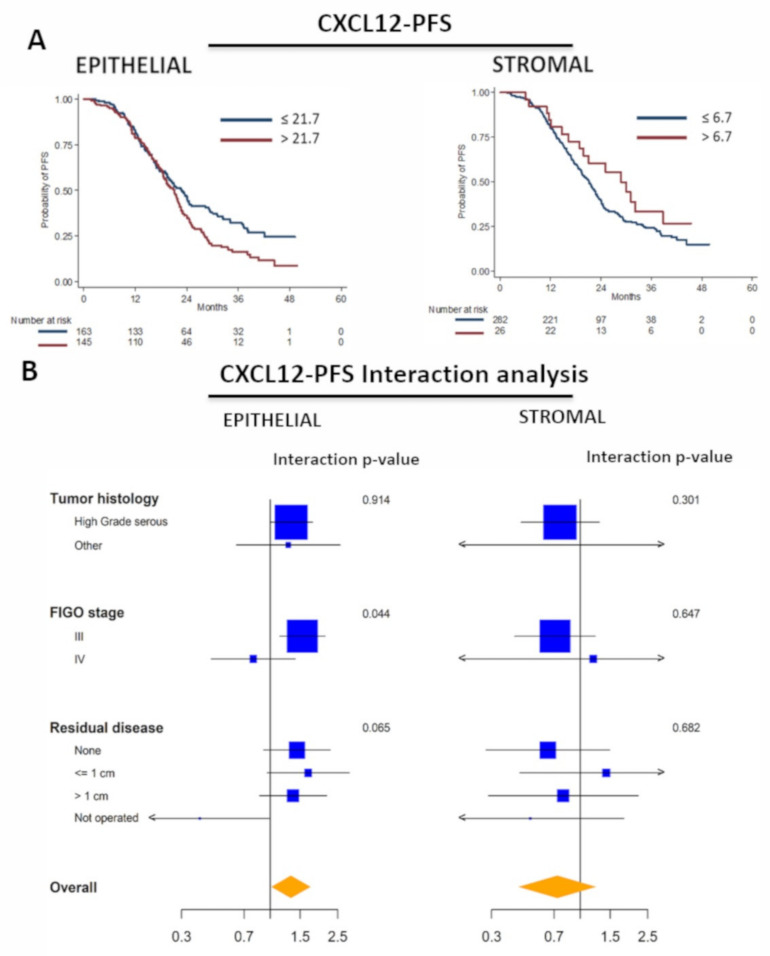
CXCL12 epithelial expression identified risk patients for PFS. PFS and epithelial and stromal CXCL12, Kaplan–Meier curves for identified cut-off on PFS CXCL12 (**A**). Forest plot of hazard ratios (HR) of progression-free survival, and 95% CI, evaluated in different subgroups (Cox proportional hazards model analysis) and interaction comparison between prognostic factor and CXCL12 in epithelial and stromal cells. The x-axis represents the odds ratio on a log scale with the reference vertical line, odds ratios (square) and 95% CI (whiskers) (**B**).

**Table 1 cancers-14-01849-t001:** Biomarkers’ co-expression based on CXCR4, CXCL12, CXCR7 IHC staining.

CXCL12	Biomarker CXCR4	CXCR7	Epithelial Cells N (%)	Stromal Cells N (%)
+	+	+	174 (56.5)	22 (7.1)
+	+	-	19 (6.2)	2 (0.6)
+	-	+	5 (1.6)	10 (3.2)
-	+	+	57 (18.5)	125 (40.6)
+	-	-	5 (1.6)	1 (0.3)
-	+	-	27 (8.8)	57 (18.5)
-	-	+	7 (2.3)	45 (14.6)
-	-	-	14 (4.5)	6 (1.9)

**Table 2 cancers-14-01849-t002:** Univariate analysis of biomarkers best cut-off for PFS and OS, original and shrunken coefficients.

	Progression Free Survival						Overall Survival					
	Original Coefficient			Shrunken Coefficients			Original Coefficient			Shrunken Coefficients		
	HR	CI(95%)	P	HR	CI(95%)	P	HR	CI(95%)	P	HR	CI(95%)	P
CXCR7 Epithelial												
>36.7	0.79	(0.6–1.04)	0.093	0.86	(0.4–1.85)	0.695	0.84	(0.56–1.28)	0.423	1.10	(0.19–6.21)	0.916
CXCR7 Stromal												
>20.0	1.16	(0.88–1.54)	0.297	1.01	(0.49–2.09)	0.974	1.23	(0.81–1.86)	0.335	0.98	(0.2–4.9)	0.985
CXCL12 Epithelial												
>21.7	1.39	(1.06–1.81)	0.016	1.31	(0.67–2.58)	0.430	1.64	(1.11–2.42)	0.014	1.51	(0.66–3.48)	0.334
CXCL12 Stromal												
>6.7	0.67	(0.4–1.11)	0.117	0.79	(0.4–1.56)	0.490	0.54	(0.24–1.24)	0.149	0.73	(0.12–4.47)	0.732
CXCR4 Epithelial												
>130.0	0.69	(0.43–1.08)	0.106	0.79	(0.34–1.83)	0.585	0.56	(0.27–1.18)	0.130	0.72	(0.13–4.02)	0.710
CXCR4 Stromal												
>65.0	0.77	(0.51–1.16)	0.209	0.91	(0.42–1.96)	0.807	0.68	(0.37–1.24)	0.210	0.87	(0.21–3.68)	0.849

## Data Availability

The datasets used and/or analyzed during the current study are available from the corresponding author on reasonable request.

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
