# Peer review of "Biological Role of Tumor/Stromal CXCR4-CXCL12-CXCR7 in MITO16A/MaNGO-OV2 Advanced Ovarian Cancer Patients"

_cancers, 2022, doi:10.3390/cancers14071849_

Round 1

Reviewer 1 Report

In this manuscript D’Alterio et al. evaluated CXCL12-CXCR4/7 expression in advanced ovarian cancer patients that have received carboplatin/paclitaxel + bevacizumab as part of a single-arm phase IV trial. They showed that CXCL12 expression in tumor cells trend towards a negative correlation with progression-free and overall survival; though this correlation falls short of statistical significance. No correlation with survival was observed in relation to CXCL12 in stroma or CXCR4/7 in tumor/stroma. Epithelial expression of CXCL12 correlates with tumor FIGO stage.

The authors well executed and controlled for the analyses to address the questions they asked. However, my concern is that wrong comparisons were performed on the data. While the authors claim that this is the first study to evaluate CXCL12-CXCR4/7 in a prospective clinical trial in advanced epithelial ovarian cancer, the impact and prognostic value of this signaling axis is well documented in numerous publications. The impact of this paper was further dampened by the fact that major findings were negative. Furthermore, support the authors’ statement that this study “strength(en) the rational for CXCL12 targeting therapy in ovarian cancer management”.

The interesting trend of differential trend of correlation between CXCL12 and PFS is an interesting lead that should be followed. Further stratification of patients should be performed to determine the prognostic value of CXCL12 in regard to its epithelial vs. stromal pattern, tumor histology, and stage, etc. Moreover, while the single arm design of the trial poses limitation, it is worth bifurcating patients as responders vs non-responders, which could yield more valuable insights instead of correlating with survival.

Author Response

Response to Reviewer 1 Comments

Reviewer #1: In this manuscript D’Alterio et al. evaluated CXCL12-CXCR4/7 expression in advanced ovarian cancer patients that have received carboplatin/paclitaxel + bevacizumab as part of a single-arm phase IV trial. They showed that CXCL12 expression in tumor cells trend towards a negative correlation with progression-free and overall survival; though this correlation falls short of statistical significance. No correlation with survival was observed in relation to CXCL12 in stroma or CXCR4/7 in tumor/stroma. Epithelial expression of CXCL12 correlates with tumor FIGO stage.The authors well executed and controlled for the analyses to address the questions they asked.

  1. However, my concern is that wrong comparisons were performed on the data. While the authors claim that this is the first study to evaluate CXCL12-CXCR4/7 in a prospective clinical trial in advanced epithelial ovarian cancer, the impact and prognostic value of this signaling axis is well documented in numerous publications. The impact of this paper was further dampened by the fact that major findings were negative. Furthermore, support the authors’ statement that this study “strength(en) the rational for CXCL12 targeting therapy in ovarian cancer management”.

We thank the reviewer for the point raised concerning the peculiarity of our study. We rephrased the sentence as follow “Among several studies (manuscript reference: ref#) ref#22 (Popple et al., 2012), ref#31 (Machelon et al., 2011), ref#32 (Pils et al., 2007) describing the crucial role of the CXCR4-CXCL12 axis in ovarian cancer this is a wide, prospective study in which the entire axis CXCR4-CXCL12-CXCR7 was systematically evaluated, in epithelial and stromal components, in a representative (N=308) population of accurately selected Stage III-IV ovarian cancer patients with adequate statistical analysis”(Discussion row:333-337). This is an added value, as the potential prognostic markers was searched within a correctly selected population on the other hand, it may be a limit as the Stage III-IV represents an advanced disease stage in which it may be hard to discriminate biological prognostic features. In accordance with other reports, the importance of targeting CXCL12 in ovarian cancer derives from the prognostic meaning of epithelial CXCL12, although not significant after application over-fitting adjustment, and the significant correlation in term of disease free survival within Stage III patients. Conclusion section was revised” … In univariate analysis epithelial high CXCL12 appeared negatively associated with PFS and OS although adjusting for overfitting the significance was lost. A significant interaction was detected in PFS analysis between epithelial CXCL12 and FIGO III stage patients. Thus, although no prognostic factors were identified through objective and rigorous statistical methods in well-controlled ovarian cancer clinical data, the evidence supports the rationale for considering CXCL12 targeting therapy in EOC management.” (Conclusion row:387-392)

  1. The interesting trend of differential trend of correlation between CXCL12 and PFS is an interesting lead that should be followed. Further stratification of patients should be performed to determine the prognostic value of CXCL12 in regard to its epithelial vs. stromal pattern, tumor histology, and stage, etc. Moreover, while the single arm design of the trial poses limitation, it is worth bifurcating patients as responders vs non-responders, which could yield more valuable insights instead of correlating with survival.

We thank the reviewer for this comment. The choice to use the two main survival outcomes (progression free survival and overall survival) as endpoints was based on data availability. As data related to PFS and OS were available for the entire analysed population, data on treatment response were available only for a fraction (63%) of the patients that retained residual disease after the main surgery, (R>0 according to the RECIST criteria). However, the median interquartile Range, (IQR) for the considered biomarkers according to response to treatment is reported in the subsequent new supplementary table 6.

Table: CXCR4, CXCL12 and CXCR7 expression and Response to treatment

Variable

Response (No)

Response (Yes)

P-Value

N = 48

N = 146

CXCR7:

    Epithelial

20 (0-60)

40 (10-76)

0.047

    Stromal

15 (0-40)

20 (0-43)

0.527

CXCL12:

    Epithelial

29 (0-83)

13 (0-56)

0.193

    Stromal

0 (0-0)

0 (0-0)

0.647

CXCR4:

    Epithelial

43 (9-67)

73 (38-93)

<0.001

    Stromal

10 (0-31)

20 (0-43)

0.072

Values are median (IQR); p-values were calculated with Wilcoxon rank test for zero-inflated data.

Interestingly high epithelial expression of CXCR4 was reported in stage III - IV ovarian cancer patients, with residual disease >1cm, responder to Carboplatin-Paclitaxel and Bevacizumab.  This is a very interesting finding as the CXCR4 axis is tightly correlated to angiogenetic pathways mainly to VEGF [manuscript refs#19 (Kryczek et al., 2005), ref #20 (Wang, So, Reierstad, & Fishman, 2006). Moreover, CXCR4 and CXCL12 increased angiogenetic factor as VEGF after bevacizumab (Liang et al., 2007), (Xu et al., 2009) in ovarian cancer patients, highlighting the cross talk between CXCR4-CXCL12 and angiogenesis (Alvarez Secord et al., 2020).  The Results and discussion section were modified as follow: Moreover, CXCR4, CXCL12 and CXCR7 expression was evaluated according to treatment response (in 194 patients, 63% of the study population) representing the patients with measurable residual disease. Interestingly higher epithelial CXCR4 expression was revealed in responders to treatment Carboplatin-Paclitaxel and Bevacizumab. (Supplementary Table 6) (results section row: 316-320) and discussed as follow: Our finding suggests that high CXCR4 at the diagnosis might identify patients, within a poor prognostic ovarian cancer patient’s category (R > 0), that will benefit from bevacizumab treatment (Ref #19). Further studies are ongoing to better categorize the angiogenetic pathway expression in the study population) (Discussion section row: 339-343).

References

Alvarez Secord, A., Bell Burdett, K., Owzar, K., Tritchler, D., Sibley, A. B., Liu, Y., Starr, M. D., Brady, J. C., Lankes, H. A., Hurwitz, H. I., Mannel, R. S., Tewari, K. S., O'Malley, D. M., Gray, H., Bakkum-Gamez, J. N., Fujiwara, K., Boente, M., Deng, W., Burger, R. A., Birrer, M. J., & Nixon, A. B. 2020. Predictive Blood-Based Biomarkers in Patients with Epithelial Ovarian Cancer Treated with Carboplatin and Paclitaxel with or without Bevacizumab: Results from GOG-0218. Clinical Cancer Research, 26(6): 1288-1296.

Kryczek, I., Lange, A., Mottram, P., Alvarez, X., Cheng, P., Hogan, M., Moons, L., Wei, S., Zou, L., Machelon, V., Emilie, D., Terrassa, M., Lackner, A., Curiel, T. J., Carmeliet, P., & Zou, W. 2005. CXCL12 and vascular endothelial growth factor synergistically induce neoangiogenesis in human ovarian cancers. Cancer Res, 65(2): 465-472.

Liang, Z., Brooks, J., Willard, M., Liang, K., Yoon, Y., Kang, S., & Shim, H. 2007. CXCR4/CXCL12 axis promotes VEGF-mediated tumor angiogenesis through Akt signaling pathway. Biochemical and biophysical research communications, 359(3): 716-722.

Machelon, V., Gaudin, F., Camilleri-Broët, S., Nasreddine, S., Bouchet-Delbos, L., Pujade-Lauraine, E., Alexandre, J., Gladieff, L., Arenzana-Seisdedos, F., Emilie, D., Prévot, S., Broët, P., & Balabanian, K. 2011. CXCL12 expression by healthy and malignant ovarian epithelial cells. BMC Cancer, 11(1): 97.

Pils, D., Pinter, A., Reibenwein, J., Alfanz, A., Horak, P., Schmid, B. C., Hefler, L., Horvat, R., Reinthaller, A., Zeillinger, R., & Krainer, M. 2007. In ovarian cancer the prognostic influence of HER2/neu is not dependent on the CXCR4/SDF-1 signalling pathway. British journal of cancer, 96(3): 485-491.

Popple, A., Durrant, L. G., Spendlove, I., Rolland, P., Scott, I. V., Deen, S., & Ramage, J. M. 2012. The chemokine, CXCL12, is an independent predictor of poor survival in ovarian cancer. British Journal of Cancer, 106(7): 1306-1313.

Wang, F. Q., So, J., Reierstad, S., & Fishman, D. A. 2006. Vascular endothelial growth factor-regulated ovarian cancer invasion and migration involves expression and activation of matrix metalloproteinases. Int J Cancer, 118(4): 879-888.

Xu, L., Duda, D. G., di Tomaso, E., Ancukiewicz, M., Chung, D. C., Lauwers, G. Y., Samuel, R., Shellito, P., Czito, B. G., Lin, P.-C., Poleski, M., Bentley, R., Clark, J. W., Willett, C. G., & Jain, R. K. 2009. Direct evidence that bevacizumab, an anti-VEGF antibody, up-regulates SDF1alpha, CXCR4, CXCL6, and neuropilin 1 in tumors from patients with rectal cancer. Cancer research, 69(20): 7905-7910.

Reviewer 2 Report

Comments:

  1. Please expand the introduction to cover more info on MITO16A and its trial.
  2. Why age of 65 is cut off age for the current study? Among 220 patients younger than 65, how many are between 18 and 30, between 30 and 45 and between and 65?
  3. The current manuscript was mainly based on IHC data, do authors have western blot analysis for CXCR4-CXCL12-CXCR7?
  4. Please add some references associated with MITO16A and its trial, such as PMID: 27993479, DOI: 10.1016/j.ygyno.2016.12.011

Author Response

Response to Reviewer 2 Comments

Reviewer #2:

Comments:

  1. Please expand the introduction to cover more info on MITO16A and its trial.

Thanks to the Reviewer MITO16A trial was widely discussed in the Introduction Section. As Follow: The MITO16A trial is a multicenter, phase IV, single arm trial. Trial aim was to explore the clinical and biological prognostic factors for advanced ovarian cancer patients receiving first-line treatment with carboplatin, paclitaxel, and bevacizumab. Patients with advanced (stage IIIB-IV) or recurrent, previously untreated, ovarian cancer received carboplatin, paclitaxel plus bevacizumab for six 3-weekly cycles followed by bevacizumab single agent until progression or unacceptable toxicity up to a maximum of 22 total cycles. Both progression-free and overall survival were used as endpoints. After a median follow-up of 32.3 months (IQR 24.1-40.4), median progression-free survival was 20.8 months (95% CI 19.1 to 22.0) and median overall survival was 41.1 months (95% CI 39.1 to 43.5). Performance status, stage, and residual disease after primary surgery were the most important clinical prognostic factors ref#25 (Daniele et al., 2021), in this context the prognostic role of epithelial and stromal expression of CXCR4-CXCL12-CXCR7 was evaluated” (introduction Row 102-114).

  1. Why age of 65 is cut off age for the current study? Among 220 patients younger than 65, how many are between 18 and 30, between 30 and 45 and between and 65?

We thank the Reviewer for the observation. For statistical methodology consistency the age cut off was the same considered in the manuscript by Daniele et al that describe the study clinical outcome ref#25 (Daniele et al., 2021). As requested, we added in the text information about prevalence of subjects in the age-classes (18-30: 1 (0.3%); 30-45: 31 (10.1%); 45-65: 188 (61.0%); >65: 88 (28.6%) (results row: 215-216)

  1. The current manuscript was mainly based on IHC data, do authors have western blot analysis for CXCR4-CXCL12-CXCR7?

This was a very valuable suggestion. The Western blot generates a signal that is proportional to the amount of the protein that exists in the sample. Unfortunately, for the present study we only had access to formalin-fixed paraffin-embedded (FFPE) samples. Nevertheless, immunohistochemistry is widely used for research and validation of biomarkers and in multicentre clinical trials as it is easy to centralize FFPE samples, have a pathology revision on the section that will then be stained and, have comparable sections to be stained for the three markers. Moreover, tumour/epithelial versus stromal were evaluated within the same samples.

  1. Please add some references associated with MITO16A and its trial, such as PMID: 27993479, DOI: 10.1016/j.ygyno.2016.12.011

As suggested by the reviewer PMID: 27993479, DOI: 10.1016/j.ygyno.2016.12.011- G. Daniele et al. Gynecol Oncol. 2017 Feb;144(2):256-259 was added to the Discussion. Although the MITO16A/MaNGO-OV2 trial intent was to explore the clinical and biological prognostic factors for advanced ovarian cancer patients receiving first-line treatment with carboplatin, paclitaxel, and bevacizumab” data on feasibility of interval debulking surgery (IDS) after carboplatin-paclitaxel-bevacizumab (CPB) ref#33 (Daniele et al., 2017) and on the absence of relevant Thromboembolic events and antithrombotic prophylaxis in advanced ovarian cancer patients treated with bevacizumab ref#34(Di Liello et al., 2021) were reported (Discussion row 344-349).

References

Daniele, G., Lorusso, D., Scambia, G., Cecere, S. C., Nicoletto, M. O., Breda, E., Colombo, N., Artioli, G., Cannella, L., Lo Re, G., Raspagliesi, F., Maltese, G., Salutari, V., Ferrandina, G., Greggi, S., Baldoni, A., Bergamini, A., Piccirillo, M. C., Tognon, G., Floriani, I., Signoriello, S., Perrone, F., & Pignata, S. 2017. Feasibility and outcome of interval debulking surgery (IDS) after carboplatin-paclitaxel-bevacizumab (CPB): A subgroup analysis of the MITO-16A-MaNGO OV2A phase 4 trial. Gynecologic Oncology, 144(2): 256-259.

Daniele, G., Raspagliesi, F., Scambia, G., Pisano, C., Colombo, N., Frezzini, S., Tognon, G., Artioli, G., Gadducci, A., Lauria, R., Ferrero, A., Cinieri, S., De Censi, A., Breda, E., Scollo, P., De Giorgi, U., Lissoni, A. A., Katsaros, D., Lorusso, D., Salutari, V., Cecere, S. C., Zaccarelli, E., Nardin, M., Bogani, G., Distefano, M., Greggi, S., Piccirillo, M. C., Fossati, R., Giannone, G., Arenare, L., Gallo, C., Perrone, F., & Pignata, S. 2021. Bevacizumab, carboplatin, and paclitaxel in the first line treatment of advanced ovarian cancer patients: the phase IV MITO-16A/MaNGO-OV2A study. International Journal of Gynecologic Cancer: ijgc-2021-002434.

Di Liello, R., Arenare, L., Raspagliesi, F., Scambia, G., Pisano, C., Colombo, N., Frezzini, S., Tognon, G., Artioli, G., Gadducci, A., Lauria, R., Ferrero, A., Cinieri, S., De Censi, A., Breda, E., Scollo, P., De Giorgi, U., Lissoni, A. A., Katsaros, D., Lorusso, D., Salutari, V., Cecere, S. C., Lapresa, M., Nardin, M., Bogani, G., Distefano, M., Greggi, S., Gargiulo, P., Schettino, C., Gallo, C., Daniele, G., Califano, D., Perrone, F., Pignata, S., & Piccirillo, M. C. 2021. Thromboembolic events and antithrombotic prophylaxis in advanced ovarian cancer patients treated with bevacizumab: secondary analysis of the phase IV MITO-16A/MaNGO-OV2A trial. International Journal of Gynecologic Cancer, 31(10): 1348.

Round 2

Reviewer 2 Report

No more comments.